# Cotton Textile with Antimicrobial Activity and Enhanced Durability Produced by *L*-Cysteine-Capped Silver Nanoparticles

**Carla Cisternas Novoa** [1,2], **Gonzalo Tortella** [2] , **Amedea B. Seabra** [3] , **María Cristina Diez** [2,4] **and Olga Rubilar** [2,4,*]

1   Doctoral Program of Science of Natural Resources, Universidad de La Frontera, Francisco Salazar 1145, Temuco 4780000, Chile; carla.cisternas@ufrontera.cl
2   Centro de Excelencia en Investigación Biotecnológica Aplicada Al Medioambiente (CIBAMA_BIOREN), Universidad de La Frontera, Francisco Salazar 1145, Temuco 4780000, Chile; gonzalo.tortella@ufrontera.cl (G.T.); cristina.diez@ufrontera.cl (M.C.D.)
3   Center for Natural and Human Sciences, Federal University of ABC (UFABC), Santo André 09210-580, Brazil; amedea.seabra@ufabc.edu.br
4   Department of Chemical Engineering, Universidad de La Frontera, Francisco Salazar 1145, Temuco 4780000, Chile
*   Correspondence: olga.rubilar@ufrontera.cl

**Abstract:** Background: In this study, L-cysteine-capped silver nanoparticles (Cys-AgNPs) were successfully linked in a cotton textile, being attached in a covalent way to the cotton fibers via esterification with the hydroxyl groups from the cellulose. The AgNPs were strongly adhered to the fiber surface through coordination bonds with the thiol groups from the *L*-cys. In addition, they were compared with biogenic silver nanoparticles produced from fungi (bio-AgNPs). Materials and methods: The characterization of the Cys-AgNP and the bio-AgNP solutions were accomplished by UV−visible (UV−Vis), Z-potential, and X-ray diffraction (XRD). After the attachment of the Cys-AgNPs and the bio-AgNPs to the raw cotton, the textile surface was characterized by variable pressure scanning electron microscopy (VP-SEM), energy dispersive X-ray (EDX), and Fourier transform infrared spectroscopy (FT-IR). The antibacterial activity was performed by disk diffusion analysis. Results: The results of the UV−Vis analysis showed the presence of AgNPs in the Cys-AgNPs and the bio-AgNPs solutions, showing the Surface Plasmon resonance (SPR) for the AgNPs among 380–420 nm. In addition, they exhibited a Z-potential of −27 and −24 mV, respectively, with the presence of elemental silver shown by the XRD analysis. The VP-SEM images from the cotton fabrics covered in Cys-AgNPs and bio-AgNPs showed the presence of spherical AgNPs on their surface, and EDX analysis revealed the presence of peaks associated with the presence of Ag, C, and O. Furthermore, FT-IR analysis exhibited peaks associated with the presence of *L*-cysteine (SH-) and carboxylic acid arising from the esterification reaction among the cellulose from cotton and the carboxylic acid in the *L*-Cys molecules. Finally, the cotton textile exhibited antibacterial activity against *Escherichia coli* and *Staphylococcus aureus*. Conclusions: This study demonstrates the ability of Cys-AgNPs to bind to the cellulose from cotton fabric so as to produce antibacterial fabrics with enhanced durability, opening a wide range of options to be further used in healthcare and other industries.

**Keywords:** cotton textile; textile modification; *L*-cysteine; antimicrobial activity; silver nanoparticles

## 1. Introduction

Over the past few years, silver nanoparticles (AgNPs) have been broadly used for their potent antimicrobial activity against several microorganisms [1]. They have been widely utilized in the healthcare, pharmaceuticals, agriculture, and environmental industries [2],

gaining ground in the development and manufacturing of new nanomaterials for different field applications [3]. In this context, the modification of cellulose fibers (cotton fabrics) with silver nanoparticles has attracted wide attention with the purpose of conferring properties such as antibacterial activity with enhanced durability [4,5]. Several methods have been used for coating AgNPs on cotton fabric, including via plasma [6,7], UV irradiation [8], sol-gel processing [9], or in situ synthesis [10], among others. To provide antibacterial activity, the metallic the silver ions are released from the cotton fabric, entering the bacterial cell, and damaging bacterial RNA and DNA, thus inhibiting replication. The slow release of the AgNPs by the binding process can confer some properties, such as sustained antibacterial activity over time; however, some loss of functionality in the cotton fabrics seems to occur after a number of laundering cycles because the conventional surface modification of textiles by AgNPs is not persistent. In addition, the involvement of chemicals in the fabrication process and the use and release into the environment of chemicals such as triclosan (and phenolic derivatives) seem to be problems to consider [11,12]. In fact, it is estimated that the polyester and cotton fiber manufacturing industry alone produces 2.7–6.4 mg of AgNPs, which are discarded in wastewater, which shows the presence of AgNPs ranging from 40 to 70 ng/L, and below 100 nm, with a few above 100 nm in the conventional effluent [13].

Nevertheless, the most-studied method is the application of binders. AgNPs can be fixed onto the cotton fiber through the chelation of hydroxyl groups from the binders with the AgNPs [14] solving these difficulties by providing steric stabilization of the AgNPs on the cotton fabrics because the surface of raw cotton presents several hydroxyl groups from the cellulose, which react with the carboxyl groups from the *L*-Cys, causing an ester bond between the agent containing the carboxyl groups and the cotton surface. In this sense, *L*-cysteine is a non-toxic, small amino acid with a carboxyl group which can react with the hydroxyl groups in the cotton surface. In addition, it possesses a thiol which, in this case, coordinates with AgNPs to form Cys-AgNPs.

In this work, we present an easy and cost-effective option for providing AgNPs coated with *L*-cysteine onto cotton fabrics. The cotton fabric samples were characterized using a scanning electron microscope (VP-SEM), Fourier transform infrared spectroscopy (FT-IR), and X-ray diffraction (XRD).

## 2. Materials and Methods

**Reagents and stock solutions:** Silver nitrate (AgNO$_3$), sodium hydroxide (NaOH), and hydroquinone (C$_6$H$_6$O$_2$) were purchased from Winkler, Santiago. Flavin adenine dinucleotide (FAD), *L*-cysteine (*L*-cys), and Pluronic F-127® were obtained from Sigma-Aldrich, St. Louis, MO, USA. To test the antimicrobial activity of the cotton fabric, Mueller–Hinton media was purchased from Merck, Darmstadt, Germany. The cotton textile (60 ends/cm, 0.42 mm thickness) was purchased from Santiago Textiles Ltd., Santiago, Chile. For all of the experimental work, solutions were prepared using analytical-grade water from a Millipore Milli-Q gradient filtration system (Millipore, 18.2 MΩ, Burlington, MA, USA). All of the reagents used in this study were of analytical grade and were used without further purification.

**Microorganisms**: *Escherichia coli* ATCC 25922 and *Staphylococcus aureus* ATCC 29213 (CLSI standard) were obtained from Applied Molecular Biology Laboratory, Universidad de La Frontera.

### 2.1. Silver Nanoparticle Synthesis and Cotton Fabric Preparation

The production of the silver nanoparticles covered in *L*-cysteine (*L*-Cys-AgNPs) was performed according to Cisternas et al., 2021, [15] using *L*-cysteine (*L*-Cys) (25 mM), hydroquinone (25 mM), flavin adenine dinucleotide (FAD) (80 nM), silver nitrate (AgNO$_3$) (1.5 mM), sodium hydroxide (NaOH) and Pluronic F-127® (2%) at a pH of 8.4 in a final volume of 25 mL. These reactives, especially hydroquinone, were used in low concentrations, biomimicking the natural properties that fungi possess to form nanostructures.

The biogenic silver nanoparticles (bio-AgNPs) were synthetized according to Rolim et al., 2019, using aqueous extract of *Stereum hirsutum*, silver nitrate (1.6 mM), and sodium hydroxide. Silver nitrate solution (1.5 mM) was used as control. Silver nanoparticles synthetized by both methods were characterized by Z Potential and dynamic light scattering (DLS). Once the AgNPs were obtained, the raw cotton fabric was prepared. For this purpose, circles with a diameter of 10 mm were cut and then washed by ultrasonic cleaning in a sodium lauryl sulfonate 2% solution for 30 min. Then, the cotton was soaked in deionized water 3 times, washed in an ethanol 80% solution for 2 h, and then rinsed with deionized water for 25 min an additional 3 times. Afterwards, the cotton textile was submerged in 3 different solutions containing cysteine-covered nanoparticles (*L*-Cys-AgNPs), biogenic nanoparticles obtained from *Stereum hirsutum* (bio-AgNPs), and $AgNO_3$ (used as a control), respectively, in a concentration of 25% v/v. All the treatments (cotton fabrics) were treated at 180 °C for 5 min, washed with 50 mL of distilled water 3 times, and dried at 50 °C in an oven for 2 h.

### 2.2. Analytical Characterization of the Cys-AgNPs, Bio-AgNPs Solutions, and the Cotton Fabric Covered in Cys-AgNPs and Bio-AgNPs

The Cys-AgNPs and bio-AgNPs solutions were analyzed by UV–Vis spectrophotometry using a Genesys10s spectrophotometer with a range of 300–700 nm to search for the characteristic SPR for AgNPs (380–420 nm). In addition, a Zetasizer Nano ZS (Malvern Instruments Co, Malvern, UK) was used to measure the average hydrodynamic diameter, the polydispersity index (PDI), and the Z Potential of the *L*-Cys-AgNPs, the bio-AgNPs, and the $AgNO_3$ solution. The experiments were performed at 25 °C using a fixed angle of 173° in disposable, folded, capillary, zeta cells, with a 10 mm path length, in an aqueous suspension. The samples were not sonicated so as to keep the original capping structure unaltered and to avoid the agglomeration of the AgNPs.

Afterwards, XRD (X-ray diffraction) of the *L*-Cys-covered fabric and the biogenic AgNPs was performed in a reflection geometry with a conventional Bruker D8 ADVANCE (CuK$\alpha$ radiation of 1.5418 Å). The samples were measured from 10° to 79° 2θ with a step size of 0.02° and a counting time of 5 s per step. In addition, the Scherrer–Debye equation was used to calculate the grain size of the AgNPs [16]. D = (k λ)/(β cos θ); where λ is the wavelength for Cukα, D is the diameter (nm), β is the full width at half-maximum (FWHM), θ is the Bragg diffraction angle, and K is a constant (9.1).

### 2.3. Characterization of the Cotton Fabric Covered in Cys-AgNPs and Bio-AgNPs

The external face composition and morphology of the cotton fabric covered in *L*-Cys-AgNPs, bio-AgNPs, and the $AgNO_3$ solution was analyzed using a VP-SEM, with a STEM SU-3500 transmission module (Hitachi-Japan). EDX analysis was determined in the STEM/EDX mode by the Ag*L* lines (Ag$L_\alpha$ at 3.000 keV and Ag$L_\alpha$ at 3.180). Finally, to view the presence of different functional groups in the sample, a FT-IR analysis was performed with a FT-IR Spectrometer Cary 600 from Agilent, using 100 mg of the air-dried sample for 24 h at 105 °C.

### 2.4. Antibacterial Activity of the Cotton Fabric Covered in L-Cys-AgNPs, bio-AgNPs, and AgNO$_3$

The antibacterial activity of the cotton textiles covered in Cys-AgNPs and the bio-AgNPs was evaluated by the disk diffusion method. For this purpose, round pieces of 10 mm were cut and sterilized with UV light for 1 h, placed on *S. aureus* and *E. coli* Muller–Hinton agar plates ($1.0 \times 10^8$ CFU/mL), and incubated at 37 °C for 24 h. The diameters of the inhibition zones (IZ) were determined.

## 3. Results and Discussions

*3.1. Characterization of the L-Cys-AgNPs and the Bio-AgNPs Previous Stabilization onto the Cotton Fabric*

Previously attached to the cotton fabric, the *L*-Cys-AgNPs, the bio-AgNPs, and the control (AgNO$_3$) were analyzed via UV−Vis, DLS, and Z Potential. Both samples exhibited a characteristic SPR for AgNPs (380–420 nm), except for the control (Figure 1), showing a maximum peak at 383 nm for *L*-Cys-AgNPs [17] and 400 for the bio-AgNPs [18]. In addition, DLS analysis showed a size of 89 nm and 109 nm, respectively, and Z Potentials of −27 mV and −24 mV (Table 1). The control (AgNO$_3$) showed no surface plasmon resonance associated with the presence of AgNPs or Z Potential.

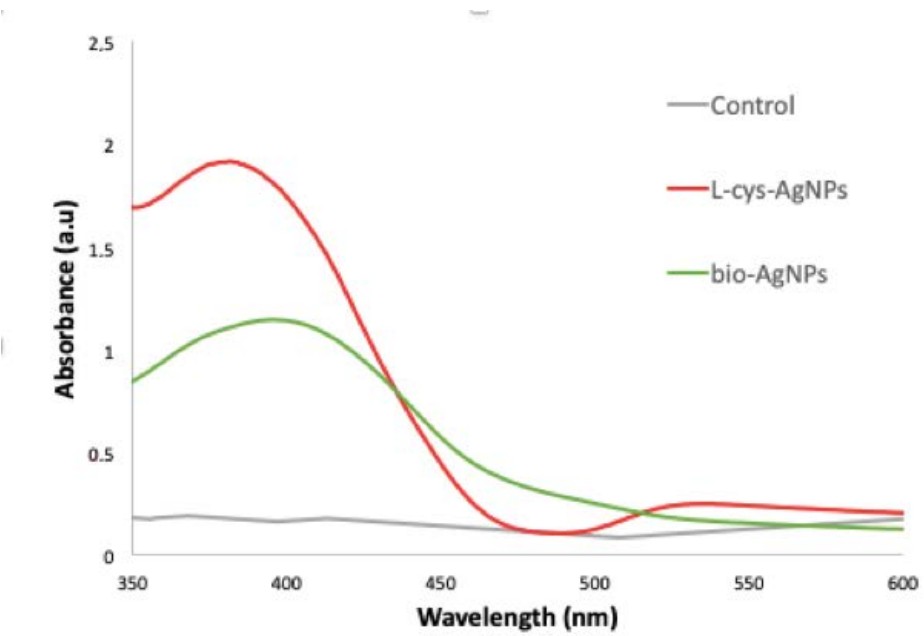

**Figure 1.** UV−Vis spectrum of the *L*-Cys-AgNPs (red), the bio-AgNPs (green), and the control AgNO$_3$ (grey).

**Table 1.** Size and Z Potential analysis of *L*-Cys-AgNPs and bio-AgNPs after 12 h of synthesis.

|  | Size (nm) | Z-Potential (mV) | PDI |
|---|---|---|---|
| *L*-**Cys-AgNPs** | 89 | −27 | 0.151 |
| **Bio-AgNPs** | 109 | −24 | 0.12 |

The XRD pattern of the cotton fabric modified with *L*-Cys-AgNPs is presented in Figure 2, showing peak values in the *L*-Cys-AgNPs of 38.2°, 44,3°, 64.2°, and 77.2°; assigned to (111), (200), (220), (311), and (222) planes, respectively, results that agree with the findings of Khan et al., 2012 [17]. These results confirm the presence of silver nanoparticles with metallic silver (Ag$^0$), in a face-centered cubic (FCC) format on the cotton fabrics covered in *L*-Cys-AgNPs. In addition, the grain size of the AgNPs was calculated using the *Debye−Scherrer* equation [19] resulting in 13.7 nm (a). On the other hand, the XRD pattern of the cotton fabric containing bio-AgNPs is not present because the concentration on the fabric was not great enough, and thus unable to produce a signal associated with the presence of AgNPs. Nevertheless, the expected outcome was the appearance of similar peaks, which were found for the *L*-Cys-AgNPs, which correspond to the presence of metallic silver.

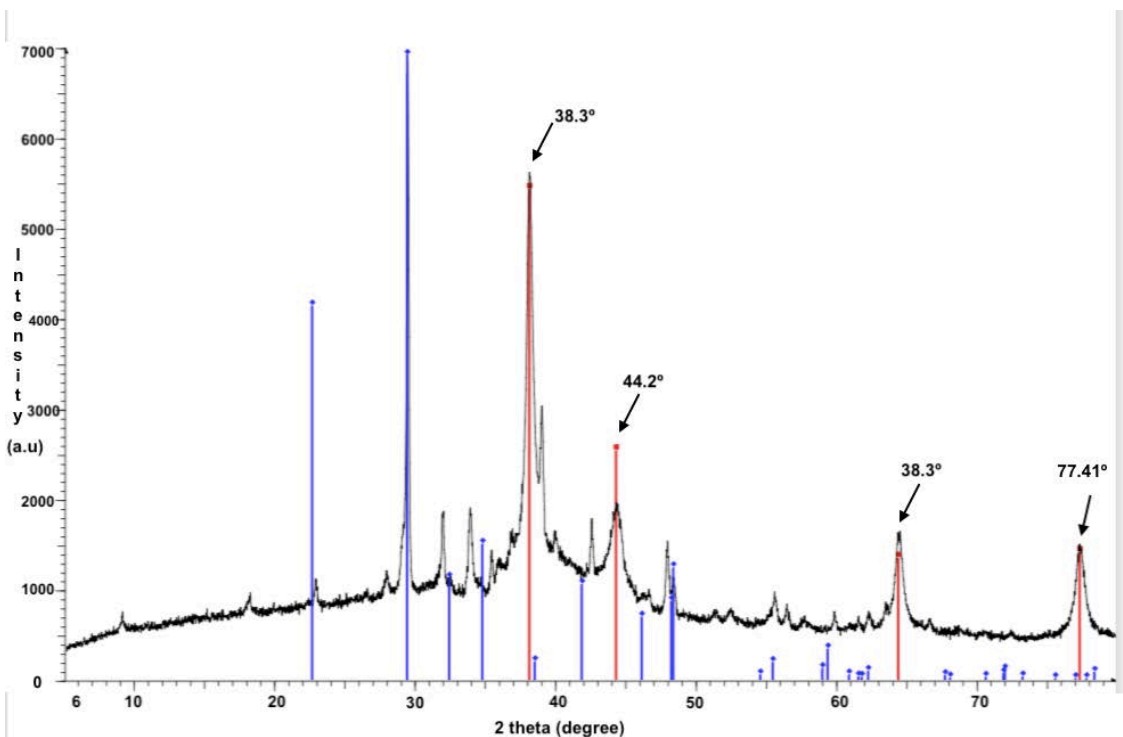

**Figure 2.** X-ray diffraction pattern of the sample containing *L*-Cys-AgNPs showing peaks at 38.3°, 44.2°, 64.5°, and 73.41°, associated with the presence of metallic silver.

### 3.2. Characterization of the Cotton Fabric Covered in L-Cys-AgNPs, Bio-AgNPs, and AgNO_3

Once the *L*-Cys-AgNPs, the bio-AgNPs, and the control were attached to the external layer of the cotton textile, the fabric was characterized by different analytical methods. The compositions of the external layer and of the modified cotton fabric were evaluated by SEM-STEM and EDX analysis (Figure 3). The results showed that the cotton fabric sample covered in *L*-Cys-AgNPs (Figure 3a,b) had abundant quasi-spherical nanoparticles, with sizes between 70 nm and 100 nm, widely distributed throughout the textile. The EDX spectrum revealed a strong peak for Ag (Figure 3f), generally showing a typical optical absorption peak approximately at 3 KeV, according to its characteristic surface plasmon resonance [20]. Additionally, EDX component analysis showed that the sample was mainly composed of carbon (47.5%) and oxygen (47.5%), which are constitutive elements of the cellulose which is present in over 90% of the cotton textile [21]. In minor concentrations, silver (2.2%) and sodium were present in the sample, arising from the presence of the *L*-Cys-AgNPs attached to the cotton and the NaOH previously used to adjust the pH, respectively.

On the other side, SEM images of the cotton fabric covered in bio-AgNPs (Figure 3c,d), showed the presence of spherical nanoparticles, with sizes ranging from 110 nm to 170 nm. In comparison with the cotton fabric covered in *L*-Cys-AgNPs, the cotton with bio-AgNPs exhibits a minor quantity of particles throughout the textile, which might be associated with a loss of the nanoparticles during the attachment process because of the lack of bonding of the bio-AgNPs, which do not chemically bond with the cellulose from the fabric. Because of this low concentration, the EDX analysis was unable to capture a signal from the sample. On the other side, Figure 4e showed the presence of dispersed aggregates of crystals of $AgNO_3$ on the fabric without the presence of AgNPs.

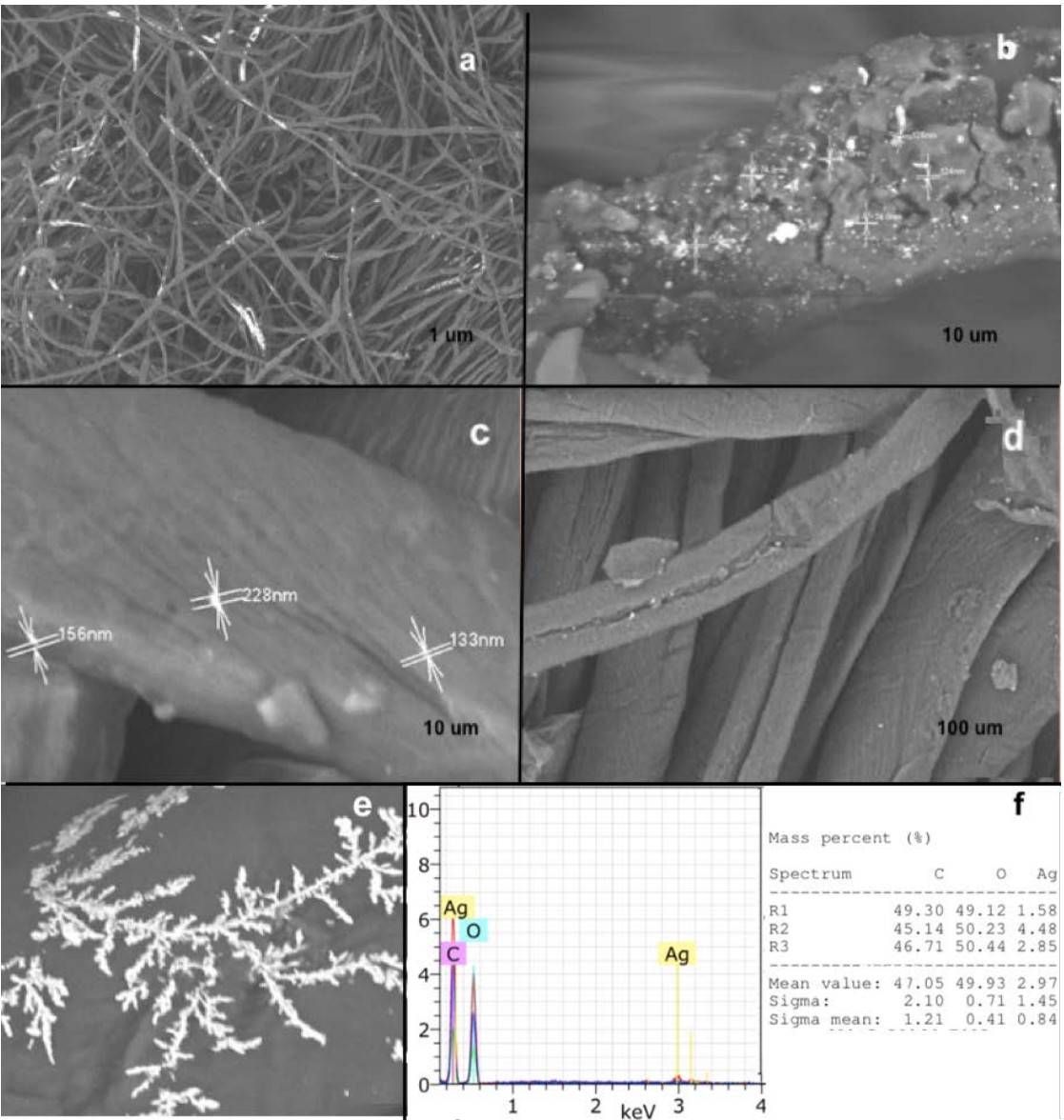

**Figure 3.** SEM-STEM analysis of the cotton fabric covered in *L*-Cys-AgNPs (**a,b**); the cotton fabric covered in bio-AgNPs (**c,d**); and the control (**e**). In addition, EDX spectra (**f**) displaying the composition of the *L*-Cys-AgNPs at pH 8.6 and the presence of a peak at 3 keV characteristic for Ag, as determined by the Ag*L*α lines.

These findings were corroborated by FT-IR, which was used to analyze the fabrics covered with *L*-Cys-AgNPs (a), bio-AgNPs (b), and AgNO$_3$ (c), by analyzing the presence of functional groups on the samples. As seen in Figure 4, the spectrum of the cotton covered with *L*-Cys-AgNPs (a) exhibited a band near 2350 cm$^{-1}$, corresponding to the presence of mercaptans -SH [22]. This band clearly disappeared for the spectrum obtained from the samples with cotton covered in bio-AgNPs (b) and the fabric covered in AgNO$_3$ (c), showing that the thiol groups from the *L*-cysteine present on the cover of the AgNPs were successfully present on the cotton. *L*-cysteine present on the AgNPs has a major role in the durability of the antimicrobial activity of the cotton, mainly because it contains a carboxylic acid (2800–3000 cm$^{-1}$) [23]. and amino group, which react with the cellulose on the cotton through an esterification reaction, being covalently bonded and preserved in time [24]. On the other side, the spectrum of the cotton covered in the bio-AgNPs (b) showed no major differences with the control (c). In this sense, these results showed a difference between the capacity of the *L*-Cys-AgNPs to form bonds with the hydroxyl groups from the cotton

in comparison with the nanoparticles synthetized with the presence of a living organism where no sign of a bonding with the cotton fabric is seen.

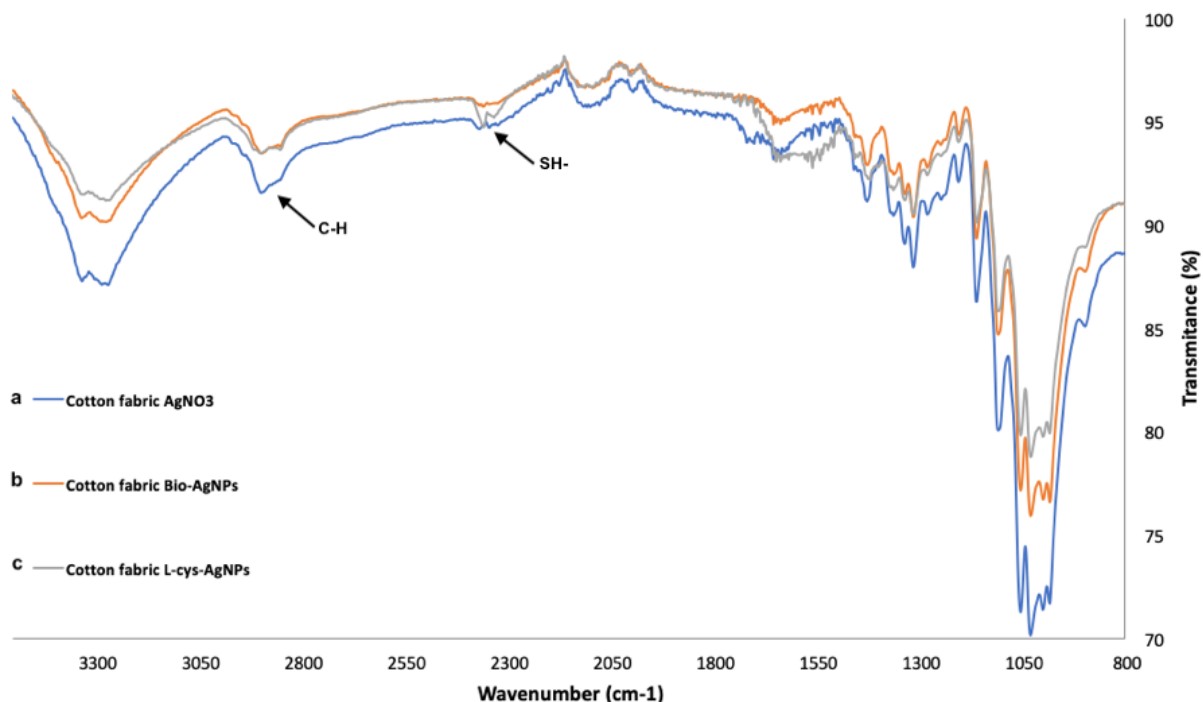

**Figure 4.** FT-IR spectra for the *L*-Cys-AgNPs (**a**), bio-AgNPs (**b**), and AgNO₃ used as control (**c**).

### 3.3. Evaluation of the Antibacterial Activity of the Optimized L-Cys-AgNPs and the Bio-AgNPs

Measurement of the antibacterial activity of the cotton fabric covered in *L*-Cys-AgNPs, bio-AgNPs, and AgNO₃ was carried out via disk diffusion assay against *Staphylococcus aureus* ATCC 14028 and *Escherichia coli* ATCC 25922 [25].

The results demonstrated that the cotton fabrics covered in *L*-Cys-AgNPs and bio-AgNPs both showed antibacterial activity against *E. coli* and *S. aureus*, (Figure 5) exhibiting an inhibition zone of 1.7 cm and 1.4 cm for *S. aureus* and 1.5 cm and 1.4 cm for *E. coli*, respectively. Nevertheless, the difference in the diameter of inhibition of both assays could be sustained by the fact that, as seen in the SEM images, cotton fabric with *L*-Cys-AgNPs exhibits a major concentration per area on the textile, a minor size of the nanoparticles, and a more negative Z Potential, avoiding aggregation. These statements are consistent with Paredes et al., 2014 [26]., showing that *L*-cysteine-covered AgNPs demonstrated antibacterial activity against MRSA (methicillin-resistant-*Staphylococcus aureus)* and *E. coli* O157:H7 strains, and even present a higher level of activity compared to commercial antibiotic or biogenic nanoparticles. On the other hand, the cotton textile covered in bio-AgNPs could have a slight antimicrobial activity, showing a minor inhibition zone on the disk because cotton fabrics covered in bio-AgNPs do not have the capability to covalently bond to the surface of the textile, losing the concentration of the bio-AgNPs in the process. As far as we know, it has been shown that the release of Ag⁺ ions is the main antibacterial mechanism of toxicity of AgNPs attaching to the surface of the cell membrane, disturbing essential functions, such as permeability and respiration. Ag⁺ ions are able to penetrate the bacteria and cause further damage, possibly by interacting with sulfur- and phosphorus-containing compounds, such as DNA [27–29].

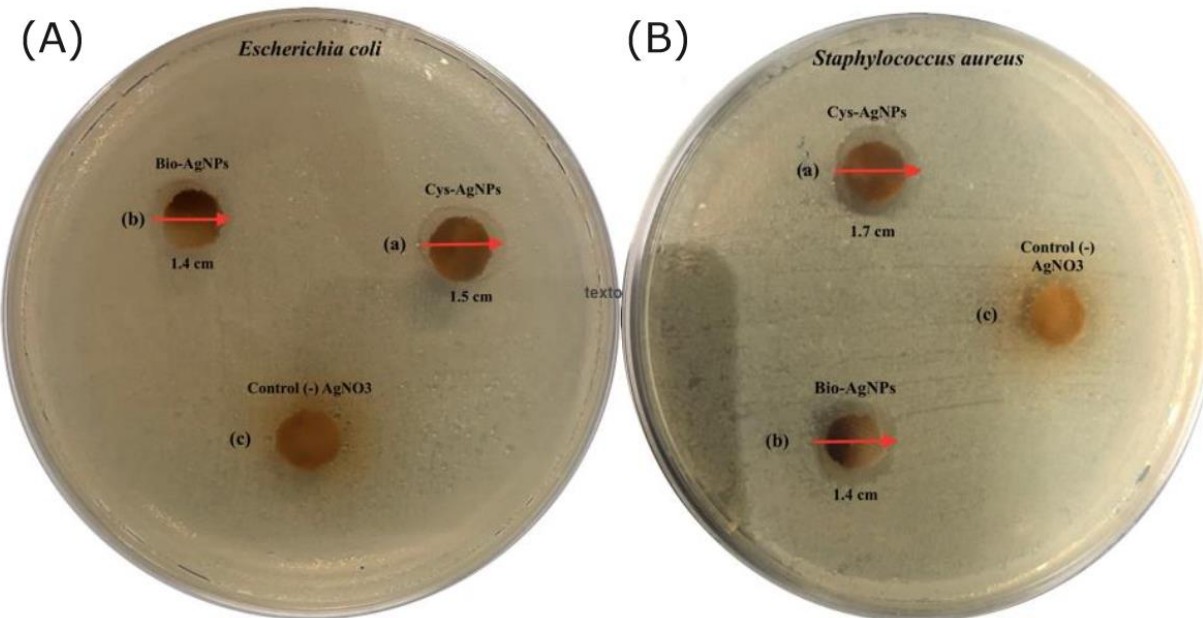

**Figure 5.** Inhibition zone test of the (**a**) Cys-AgNP cotton, (**b**) bio-AgNP cotton. and (**c**) control (-) AgNO$_3$ in *Escherichia coli* (**A**) and *Staphylococcus aureus* (**B**).

## 4. Conclusions

The novelty of this work is that we use a biomimetic approach in an easy and affordable way, with the use of *L*-cysteine-covered nanoparticles produced by fungal metabolites from free-mycelium extracellular medium, and only using steric bonds for the attachment of the AgNPs to the cotton surface. This is caused by the *L*-cys moieties enhancing the adhesion capability of the AgNPs in comparison with the bio-AgNPs, which was demonstrated by the analytical characterization of the textiles. In addition, the *L*-Cys-AgNPs are reduced and ready to use, leaving toxic and non-biocompatible components used in other methods to reduce the silver ions and turn them into nanoparticles. In addition, the antibacterial activity of the cotton fabric covered in *L*-Cys-AgNPs and the bio-AgNPs was demonstrated against *Escherichia coli* and *Staphylococcus aureus*; however, *L*-Cys-AgNP cotton fabric has advantages, such as the significant durability of the antibacterial activity on the fabrics. For future research, this is the first step towards producing wound-healing dressings that are biocompatible with humans and the healthcare industry. Further studies regarding the mechanisms and the foundations of the bonding among the cotton fabric and the *L*-Cys-AgNPs are necessary, showing a potential use in biomedical applications

**Author Contributions:** Conceptualization, C.C.N. and O.R., methodology, C.C.N. and G.T.; software, C.C.N.; validation, A.B.S. and O.R.; formal analysis, C.C.N.; investigation, C.C.N.; resources, M.C.D. and O.R.; data curation, G.T.; writing—original draft preparation, C.C.N.; writing—review and editing, C.C.N., M.C.D., G.T., A.B.S. and O.R.; visualization, G.T.; supervision, O.R., project administration, O.R.; funding acquisition, M.C.D. and O.R. All authors have read and agreed to the published version of the manuscript."

**Funding:** This research was funded by FONDECYT 1191089, ANID/FONDAP/15130015, CONYCIT-FAPESP (2018/08194-2), DI20-1003 project, CONICYT-REDES 180003, CNPq (404815/2018-9).

**Institutional Review Board Statement:** Not applicable.

**Informed Consent Statement:** Not applicable.

**Data Availability Statement:** The study did not report any further data.

**Acknowledgments:** We appreciated the support of FONDECYT 1191089, ANID/FONDAP/15130015, ANID-FAPESP (2018/08194-2), ANID-REDES 180003, and CNPq (404815/2018-9).

**Conflicts of Interest:** The authors declare no conflict of interest.

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
