# Peer review of "Cotton Textile with Antimicrobial Activity and Enhanced Durability Produced by L-Cysteine-Capped Silver Nanoparticles"

_processes, doi:10.3390/pr10050958_

Round 1
Reviewer 1 Report
In this work, the authors show the functionalization of cotton fabric using AgNPs in two presentation, on the one hand binding them with L-cysteine, on the other hand using bio-AgNPs.
The paper is well written, although the introduction and discussions should be revisited thoroughly by adding several previous works and making the contrast of the properties with the present one. Otherwise, the report looks like a communication, more than an article.
Aside from that, the use of hydroquinone is not that “environmentally friendly”.
The main question I would have and that the authors should address is: How is that the bio-AgNPs were not binded with L-cysteine also. It should be natural to think that the bio-AgNPs did not work because they did not adsorb well, thus, the most innovative part of the paper could be adding bio-AgNPs to cotton effectively.
Introduction:
The authors should give a better account of the previous literature, quantitys of AgNPs that go into water sources in the washing cycles, and additional information on the adsorption of AgNPs on cotton with natural binders.
Zhang, S., Zhang, T., He, J. et al. Effect of AgNP distribution on the cotton fiber on the durability of antibacterial cotton fabrics. Cellulose 28, 9489–9504 (2021). https://doi.org/10.1007/s10570-021-04113-0
Facile approach for large-scale production of metal and metal oxide nanoparticles and preparation of antibacterial cotton pads https://doi.org/10.1016/j.carbpol.2017.01.059
Investigation of the antimicrobial and wound healing properties of silver nanoparticle-loaded cotton prepared using silver carbamate Tae-Sung Kim, Jae-Ryung Cha, Myong-Seon Gong https://doi.org/10.1177/0040517516688630
ÄŒuk, N., Šala, M. & Gorjanc, M. Development of antibacterial and UV protective cotton fabrics using plant food waste and alien invasive plant extracts as reducing agents for the in-situ synthesis of silver nanoparticles. Cellulose 28, 3215–3233 (2021). https://doi.org/10.1007/s10570-021-03715-y
Xu, S., Zhang, F., Yao, L. et al. Eco-friendly fabrication of antibacterial cotton fibers by the cooperative self-assembly of hyperbranched poly(amidoamine)- and hyperbranched poly(amine-ester)-functionalized silver nanoparticles. Cellulose 24, 1493–1509 (2017). https://doi.org/10.1007/s10570-016-1178-5
Multifunctional organic cotton fabric based on silver nanoparticles green synthesized from sodium alginate Sakil Mahmud, Nahid Pervez, Muhammad Abu Taher, https://doi.org/10.1177/0040517519887532
Andra, S., Balu, S.k., Jeevanandam, J. et al. Surface cationization of cellulose to enhance durable antibacterial finish in phytosynthesized silver nanoparticle treated cotton fabric. Cellulose 28, 5895–5910 (2021). https://doi.org/10.1007/s10570-021-03846-2
Also the previous research with L-cysteine, similarities and differences with this research, and what would be the originality, and the scope of the present work (e.g., which is the novelty)
Xu, Q., Ke, X., Cai, D. et al. Silver-based, single-sided antibacterial cotton fabrics with improved durability via an L-cysteine binding effect. Cellulose 25, 2129–2141 (2018). https://doi.org/10.1007/s10570-018-1689-3
Enhancing the surface affinity with silver nano-particles for antibacterial cotton fabric by coating carboxymethyl chitosan and l-cysteine https://doi.org/10.1016/j.apsusc.2019.143673
Xu Q et al (2019) Enhancing the surface affinity with silver nano-particles for antibacterial cotton fabric by coating carboxymethyl chitosan and l-cysteine. Appl Surf Sci. https://doi.org/10.1016/j.apsusc.2019.143673
Please change the reference style throughout the manuscript [1]: Cisternas 2021 is not in the references
Results:
On the other side, XRD pattern of the cotton fabric containing bio-AgNPs is not present: Is there an explanation why???
Also, EDX Spectrum (f) displaying the composition of the L-cys-AgNPs at pH 8.6 and the presence of a peak at 3 keV characteristic for Ag element, determined by the AgLα lines. : Authors should present also here in the figure why the bio-AgNPs EDX were not shown (it is in the text, although here would be good to complete the figure data)
huge potential applications: Please change this statement
Best regards
Reviewer 2 Report
Dear Authors,
This study which demonstrates the ability of the Cys-AgNPs to bind the cellulose from the cotton fabric, to produce antibacterial fabrics with enhanced durability, is very interesting approach to obtaining a wide range of options to be further used in the healthcare and industry.
Please, check the paper and correct some technical errors.
Also, please try to replace SEM graphs in figure 3 with more clear ones, this one is not a good quality.
In Figure 4 please add a characteristic peaks after different treatments for checking a diferencies.
Maybe, in next paper we can exept durability test of such treated AgNPs particles and L-cys-AgNPs cotton materials.
With kind regards,
Reviewer

Reviewer 3 Report
The manuscript presents an interesting study. The title is correct and clear, and the summary is well-ordered and compressive. I do not have any consideration in this section.
However it does have shortcomings that need to be addressed before any further considerations should be made.
First, the use of English needs to be corrected. There are far many grammatical and style errors throughout the text.
The introduction section needs to be improved in relation with existing already studies and up-to-date references.
In general, the results and the discussion are well expressed.
Conclusion: This section needs to be better explained. However, there will be no future studies? I recommend the authors put de something adds two sentences referring them to future studies.
I recommend that authors follow the same format for all references.
Round 2
Reviewer 1 Report
The authors should not only answer the following statements in the letter. They should be incorporated inside the manuscript. Otherwise, the reader would have several questions unanswered:
- Aside from that, the use of hydroquinone is not that “environmentally friendly”.
R: In this sense, the use of hydroquinone is based on the metabolites found in the free-mycelium extracellular medium, where hydroquinone acts as an electron donor (Duran et al. 2005, Kumar et al, 2007). Agreeing with this, the use of hydroquinone in this context is used in low concentrations, biomimicking the natural properties that fungi possess to form nanoestructures. However, we would be pleased to remove any reference that mentions that hydroquinone is environmentally friendly.
- The main question I would have and that the authors should address is: How is that the bio-AgNPs were not binded with L-cysteine also. It should be natural to think that the bio-AgNPs did not work because they did not adsorb well, thus, the most innovative part of the paper could be adding bio-AgNPs to cotton effectively.
R: First, thanks for this question, it is a very good observation. However, the address on this paper was to show a clear difference between the capacity of the L-cys-AgNPs to form bonds with the hydroxyl groups from the cotton, and compare them with the nanoparticles synthetized with the presence of a living organism. On the other side, we think for a next work is a very approachable direction that we would be interested to follow.
- Also the previous research with L-cysteine, similarities and differences with this research, and what would be the originality, and the scope of the present work (e.g., which is the novelty)
R: The novelty of this work is that we use a biomimetic approach, with the use of L-cysteine covered nanoparticles produced by fungal metabolites from free-mycelium extracellular medium and only using steric bonds for the attatchment of the AgNPs to the cotton surface. Also, the l-cys-AgNPs are reduced and ready to use, leaving toxic and non-biocompatible components used in other works to reduce the silver ions and turn them into nanoparticles. This is the first step to produce wound healing dresses that are biocompatible with human and the healthcare industry.
Results:
- On the other side, XRD pattern of the cotton fabric containing bio-AgNPs is not present: Is there an explanation why???
R: In spite we sent the sample for the analysis of the XRD pattern in the cotton fabric containing Bio-AgNPs, the concentration on the fabric was not enough, being unable to produce peaks associated with the presence of AgNPs.
- Also, EDX Spectrum (f) displaying the composition of the L-cys-AgNPs at pH 8.6 and the presence of a peak at 3 keV characteristic for Ag element, determined by the AgLα lines: Authors should present also here in the figure why the bio-AgNPs EDX were not shown (it is in the text, although here would be good to complete the figure data)
R: The EDX spectrum for the cotton covered in Bio-AgNPs was made, however, the concentration of the Bio-AgNPs was not enough to produce a signal in the EDX pattern so we decided not to show it.
Best regards
Author Response
Thanks a lot for your commentaries. There were very richful.
I really appreciate
All of the answers were wrote down on the manuscript.
